# A Comparison of Laboratory and Industrial Processes Reveals the Effect of Dwell Time and UV Pre-Exposure on the Behavior of Two Polymers in a Disintegration Trial

**DOI:** 10.3390/polym16121650

**Published:** 2024-06-11

**Authors:** Simon Schick, Robert Groten, Andreas Weinberger, Gunnar H. Seide

**Affiliations:** 1Aachen-Maastricht Institute for Biobased Materials (AMIBM), Faculty of Science and Engineering, Maastricht University, Brightlands Chemelot Campus, Urmonderbaan 22, 6167RD Geleen, The Netherlands; s.schick@maastrichtuniversity.nl; 2Department of Textile and Clothing Technology, Niederrhein University of Applied Sciences, Campus Mönchengladbach, Webschulstrasse 31, 41065 Mönchengladbach, Germany; 3IFG ASOTA GmbH, Schachermayerstrasse 22, 4020 Linz, Austria

**Keywords:** biopolymers, biodegradable fiber, melt spinning, polybutylene succinate, polylactic acid, spinning scale, multi-step degradation

## Abstract

Biodegradable biopolymers such as polylactic acid and polybutylene succinate are sustainable alternatives to traditional petroleum-based plastics. However, the factors affecting their degradation must be characterized in detail to enable successful utilization. Here we compared the extruder dwell time at three different melt-spinning scales and its influence on the degradation of both polymers. The melt temperature was the same for all three processes, but the shear stress and dwell time were key differences, with the latter being the easiest to measure. Accelerated degradation tests, including quick weathering and disintegration, were used to evaluate the influence of dwell time on the structural, mechanical, and thermal properties of the resulting fibers. We found that longer dwell times accelerated degradation. Quick weathering by UV pre-exposure before the disintegration trial, however, had a more significant effect than dwell time, indicating that degradation studies with virgin material in a laboratory-scale setting only show the theoretical behavior of a product in the laboratory. A weathered fiber from an industrial-scale spinning line more accurately predicts the behavior of a product placed on the market before ending up in the environment. This highlights the importance of optimizing process parameters such as the dwell time to adapt the degradability of biopolymers for specific applications and environmental requirements. By gaining a deeper insight into the relationship between manufacturing processes and fiber degradability, products can be adapted to meet suitable performance criteria for different applications.

## 1. Introduction 

Polylactic acid (PLA) and polybutylene succinate (PBS) are versatile biopolymers with many environmental benefits [1,2]. PLA has outstanding mechanical stiffness and strength [3] and also breaks down under industrial conditions (58 °C) [4]. It is therefore suitable for biomedical, agricultural, textile, and food packaging applications [1,2]. Conversely, PBS breaks down at lower temperatures (28 °C) [4], making it a sustainable alternative to polyethylene terephthalate and polypropylene (which have similar properties), and potentially suitable for home composting [5]. Whereas conventional plastics derived from fossil resources persist in the environment [6], PLA and PBS are derived from biobased raw materials, so biodegradation is a sustainable end-of-life scenario, particularly if disposal occurs close to the site of use, thus reducing the impact of logistics [7,8]. The demand for environmentally friendly products and packaging solutions has driven the development and applications of PLA and PBS [9]. However, the market share of both polymers is influenced by the cost of raw materials [10]. The PLA market share was 18.9% in 2021, and the PBS market share is expected to reach 16% by 2026 [11]. One interesting application is the melt binding of fibers in teabags, which would reduce the 9000 tons of persistent waste generated by the disposal of 30 billion used teabags per year in Germany [4]. 

To support the transition to biobased and biodegradable plastics, it is necessary to assess the environmental performance of polymers throughout their life cycle (Figure 1). This is often achieved by life cycle assessment (LCA), but such studies require the availability of high-quality data [12,13]. This is difficult to achieve for new and emerging technologies because, even if data are available, the technological readiness level (TRL) tends to be too low for comparison to established technologies [14]. There is also a lack of primary data modeling end-of-life scenarios, such as home composting [15]. It is therefore necessary to build reliable datasets that identify the real environmental benefits of biopolymers such as PLA and PBS.

Like their petrochemical counterparts, biopolymers processed by melt spinning experience thermomechanical stress that causes their degradation, the severity of which depends on the temperature, shear stress, and dwell time [16,17,18]. The temperature can be maintained when scaling up, but shear stress can only by modeled. The dwell time of the polymer in the extruder can be measured precisely at different process scales, therefore offering an excellent parameter for comparison. 

PLA has been subjected to accelerated aging in extrusion and injection molding studies [19], and different molecular weights of PLA have been tested by hydrolysis [20]. PLA fibers have also been compared to cellulose fibers, such as hemp, jute and sisal, for their degradation in farmland soil for up to 11 days under controlled conditions [21]. Poly(butylene succinate-co-butylene adipate) samples melt blended with TiO_2_ have been tested by UV degradation [22], and PBS prepared by injection molding has been compared to low-density polyethylene by accelerated weathering [23]. PBS mixed with Mg–Al was prepared by melt compounding and tested in a marine degradation setting [24]. PLA blended with various polymers has been tested by accelerated aging [25], and PBS in a composite with jute has been tested in a biodegradation setting [26]. Blends of PLA and PBS have also been investigated in disintegration studies with inorganic fillers such as talc and chalk [27]. Poly(3-hydroxybutyrate-co-3-hydroxyhexanoate) in medical devices has been tested for hydrolysis [28]. Polymers are also prone to degrade during processing [16,29]. Even broader studies of biodegradable man-made fibers look only at burial trials in natural soil or compost for PLA, polyhydroxyalkanoate-PLA blends, or bi-component fibers prepared from PBS and PLA [30]. 

The studies above examined individual polymers and their degradability in detail. Multiple grades, polymer blends, or different polymers are often evaluated using a single degradation mechanism. The possibility of pre-exposure causing a second degradation mechanism is disregarded in studies focusing on one degradation mechanism alone. In addition, the influence of different production scales (and therefore differences in dwell time) has been overlooked. The relationship between dwell time and degradation in two consecutively tested degradation mechanisms can be clearly defined as a knowledge gap. We therefore investigated the influence of dwell time during the processing, accelerated weathering, and disintegration of PLA and PBS. Three process scales were identified with different dwell times: laboratory, pilot, and industrial scales. We also examined the influence of accelerated weathering on disintegration to demonstrate the realistic degradation of biopolymer products that end up in the environment due to accidental or deliberate improper disposal. We focused on the relationship between the laboratory and industrial fibers in virgin and pre-exposed settings, thus contributing useful data for future LCA studies.

## 2. Materials and Methods

### 2.1. Materials and Samples

PLA 6202D and PBS FZ91PM granular polymers, as well as fibers produced at three different scales, were provided by IFG ASOTA GmbH Linz (Linz, Austria). The fibers were given codes referring to the polymer (PLA or PBS) followed by an underscore and then the scale (LAB for laboratory, SF for staple force representing the SF1000 pilot line, and IND for the industrial line), followed by another underscore and then the sample type according to the degradation pathway (Figure 2). Pathway 2.1 proceeds from the granular polymer and fiber to fully disintegrated samples (ISO) and is represented by orange highlighting in subsequent tables. Pathway 2.2 proceeds from the granular polymer and fiber to UV exposure (300h) and a disintegrated sample with UV pre-loading (300h + ISO), represented by blue highlighting in subsequent tables. A PLA geotextile (PLA_GEO) was used as a comparison. This geotextile is provided by IFG ASOTA GmbH Linz (Linz, Austria) and is constructed from IFG ASOTA (Linz, Austria) fibers that are already on the market.

### 2.2. Differential Scanning Calorimetry (DSC)

A Mettler Toledo polymer DSC device was used to determine the melting temperature (T_m_) and melt enthalpy (ΔH_m_) in three cycles of heating–cooling–heating. The first heating cycle was set from 25 to 220 °C, the cooling cycle was set from 220 to −30 °C, and the second heating cycle from −30 to 220 °C, with a rate of 10 °C/min under a constant stream of nitrogen at 50 mL/min. The data were analyzed using Mettler Toledo STARe software (version V16.20c). Melt enthalpy values for 100% crystalline polymers (ΔH_m_100) were taken from the literature. The crystallinity (X_C_) achieved during spinning was then calculated using Equation (1) [16], where ΔH_m_ is the melt enthalpy from the first heating cycle and ΔH_m_100 is the literature value for the 100% crystalline polymer (93.7 J/g for PLA and 110.3 J/g for PBS) [31]. The PLA and PBS samples are shown in red and blue, respectively:(1)$$X_{C}=\frac{{∆H}_{m}}{{∆H}_{m}^{100}}\times 100$$

### 2.3. Fourier Transform Infrared (FTIR) Spectroscopy

We determined the chemical profile of the granular polymer, fibers before and after UV exposure, and fibers following the disintegration and UV + disintegration trials using an Alpha 2 FTIR device (Bruker, Billeric, MA, USA) with a single diamond ATR system. Values were recorded in the range of 4000–500 cm^−1^ with a resolution of 16 cm^−1^. We normalized 21 scans to ensure comparability between degradation stages. The relationship between the reference and selected peaks was calculated using Equation (2) [32]:(2)$$\mathrm{Peak}\;\mathrm{height}\;\mathrm{ratio}=\frac{{\mathrm{Height}}_{\mathrm{Peak}\;X}}{{\mathrm{Height}}_{\mathrm{Reference}\;\mathrm{Peak}}}$$

Changes in different chemical groups (carbonyl, hydroxyl, and vinyl) were calculated using the indices in Equations (3)–(5), respectively [33,34]:(3)$$\mathrm{Carbonyl}\;\mathrm{Index}\;\left(CI\right)=\frac{I_{1710-1713}}{I_{2945}}$$
(4)$$\mathrm{Hydroxyl}\;\mathrm{Index}\;\left(HI\right)=\frac{I_{3423-3429}}{I_{2945}}$$
(5)$$\mathrm{Vinyl}\;\mathrm{Index}\;\left(VI\right)=\frac{I_{917}}{I_{2945}}$$

### 2.4. Gel Permeation Chromatography (GPC)

The number average molecular weight (M_n_) of each polymer was determined by GPC using a 1260 Infinity System (Agilent Technologies, Santa Clara, CA, USA). The PBS samples were dissolved in hexafluoro-2-isopropanol (HFIP) containing 0.19% sodium trifluoroacetate and were separated at a flow rate of 0.33 mL/min. The PLA samples were dissolved in chloroform and tested externally. GPC analysis was carried out at each processing step (Figure 2). The *Mn* was used to calculate the degradation parameter *K* (Equation (6) [16]). *K* is the molecular weight ratio before and after polymer degradation and is used evaluate the influence of different degradation steps [16]: (6)$$K=\frac{M_{n}\;\mathrm{unprocessed}}{M_{n}\;\mathrm{processed}}$$

### 2.5. Tensile Test

Fiber diameter was determined using a Vibroskop-500, and the tensile force and elongation were determined using a Vibrodyn-500 (both from Lenzing Instruments, Gampern, Austria). We used a clamping length of 20 mm and applied a titer-specific pre-tension (titer × 0.7 mg) and a test speed of 80 mm/min. Both tests were based on DIN EN ISO 5079 [35]. 

### 2.6. Fiber Preparation

Fiber samples were prepared using laboratory spinning equipment (LAB), the SF1000 pilot line (SF), or the industrial line (IND), using the parameters shown in Table 1. The dwell time was measured with a stopwatch, which was started when the pigmented granulate was added to the extruder and stopped when the first colored fibers emerged from the spinning plate. The dwell time was measured five times and rounded to the next full minute. The settings at each scale were selected to favor a stable spinning process and high-quality fibers. For the SF and IND lines, the process was stable over several hours.

After mechanical testing, the fibers were cut and converted into a nonwoven. This involved manual cutting of the LAB samples and the in-production cutting and carding of 100 g samples at the other scales (two rounds in a card clothing machine). After aligning the fibers, the nonwoven was solidified by needle punching. Needle Machine 1 (needle density per meter = 1100) was applied once from the top and once from the bottom before the finishing pass on Needle Machine 2 (needle density per meter = 3000) using a pattern that favored fiber integration and easy handling (Figure 3). After solidification, the nonwoven fabrics were cut into 100 cm^2^ samples for FTIR analysis and UV exposure.

### 2.7. Quick Weathering

Accelerated weathering was applied to duplicate samples using a Q-Sun XE-2HS device (Q-Lab). We tested the full UV range, with each test comprising three cycles lasting a total of 12 h to simulate summer sunlight at noon followed by rain and summer night. Accordingly, the first cycle was 8 h of UV exposure (60 W/m^2^) at 50% humidity and 38 °C. The second cycle involved spraying the samples with deionized water at 38 °C while exposing them to UV (60 W/m^2^) for 15 min. The third cycle was a dark period of 3 h 45 min at 95% humidity and 38 °C. The cycles were repeated until the total UV exposure was 300h. The overall loading of the samples with UV radiation was similar to the quick weathering of plastics under DIN standard EN ISO 11,341 [36] (light cycle), ISO 4892-2 [37] (spraying), or ISO 4892.3 [38] and simulated exposure for several months/years in days [39]. 

### 2.8. Laboratory-Scale Disintegration

Disintegration was carried out in close alignment with DIN standard EN ISO 20200. Artificial waste comprising 40% sawdust, 30% rabbit feed, 10% mature compost (ARGE Kompost Enns), 10% corn starch, 5% sucrose, 4% mouse germ oil, and 1% urea was mixed in a tumbler mixer for several minutes. The water content was adjusted with distilled water so that the water did not pour out when squeezed by hand. The nonwoven samples were cut into 25 × 25 mm pieces and dried at 40 °C for 12 h before they were weighed, soaked in distilled water for 30 s, mixed with the artificial waste to a final proportion of 2% (*w*/*w*), and filled into 0.5 L plastic containers with lids to prevent drying. The containers were weighed before incubating at 38 °C (±2 °C) for 45 days. Twice a week, the containers were weighed, brought to the initial weight with distilled water, and mixed manually with a spatula. After 45 days, the containers were opened, and the waste was dried at 50 °C. Bigger lumps were carefully crumbled by hand. After drying, the waste was sieved according to ISO 3310-1 [40] (mesh size 10, 5, and 2 mm), and the retained fragments were freed from compost [41]. 

## 3. Results

### 3.1. Mechanical Properties of the Spun Fibers

We compared the mechanical properties of 10 single fibers spun at each of the three scales, revealing a clear distinction between PLA and PBS (Table 2). PLA showed higher values for tenacity (except PBS_LAB), elongation, and crystallinity. PLA_IND achieved the highest tenacity overall (26.96 cN/tex), far exceeding that of PBS_IND (15.97 cN/tex). The elongation of PBS_IND (214.74%) was almost five-fold that of PLA_IND (45.66%). PBS_IND was lower in crystallinity (51.46%) than PBS_SF (64.51%) and PBS_LAB (54.3%), whereas PLA_IND was higher in crystallinity (39.88%) than the other scales, with PLA_SF being the lowest (33.84%). The tenacity of PLA fibers increased with scale, whereas PBS_LAB showed the highest tenacity among the PBS fibers (23.41 cN/tex) and PBS_SF the lowest (12.86 cN/tex). There was no correlation between scale and crystallinity, with the pilot line producing the highest value for PBS (and overall) but the lowest for PLA (and overall). 

### 3.2. Change Due to Degradation

Two of the samples (PLA_LAB and PLA_SF) disintegrated completely during the trials and could not be tested further, so no post-degradation DSC, FTIR, or GPC data are presented in the following figures and tables. For the PLA_GEO sample, PLA_Granulate was used for comparability. 

#### 3.2.1. Differential Scanning Calorimetry

The PLA and PBS samples behaved differently in the DSC experiments, but all samples of the same polymer, regardless of the production scale, gave similar results. Representative DSC curves of the first heating cycle are therefore shown as examples in Figure 4 (PLA_IND) and Figure 5 (PBS_IND). A significant change in the melt enthalpy, which is used to calculate crystallinity, was observed when comparing the curves during the degradation process. Although the PLA_IND surface changed significantly in terms of enthalpy and shape, the PBS_IND surface only changed in terms of enthalpy when comparing the nonwoven to the 300h + ISO treatment.

The crystallinity changed significantly during degradation path 2.1 (Figure 6) and degradation path 2.2 (Figure 7). PBS showed little change between the granulated and ISO samples in the absence of UV exposure, whereas a slight increase in crystallinity was observed for PLA_IND (Figure 6). However, UV pre-exposure resulted in significant changes (Figure 7). The crystallinity of the PBS_SF sample increased almost linearly during degradation, whereas the other PBS samples showed a decline from the granulated sample to the fiber and 300h UV exposure before rising again. The crystallinity of the PLA samples was not significantly affected by spinning, but increased with UV exposure and during degradation. The ISO disintegration at the end reduced the crystallinity of these samples again. The PLA_GEO sample showed no change due to UV exposure, and the decrease in crystallinity was similar to that of PLA_IND. The crystallinity values are listed in Table 3.

We also observed changes in the T_m_ during degradation without prior UV exposure (Figure 8) and with UV pre-exposure (Figure 9). The T_m_ change for PBS was marginal, with all samples differing by only 1–2 °C during degradation, but the change for PLA was more significant. The melt temperature of the PLA fiber was higher than that of the granular PLA sample at all three scales. The most significant change in T_m_ was a drop to ~150 °C for PLA_IND, which was more severe without UV exposure (Figure 8) but also occurred following UV exposure (Figure 9). For PLA_GEO, we observed a slight rise followed by a steep drop in T_m_ during degradation, but the drop was not as severe as that observed for PLA_IND.

#### 3.2.2. FTIR Spectroscopy

The change measured by FTIR was based on the CI (Table 4), HI (Table 5), and VI (Table 6). During the degradation process, all three indices decreased slightly for the PBS samples and more sharply for the PLA samples. Furthermore, the changes in all three indices were more significant following pre-exposure to UV (pathway 2.2). For PLA, all samples with UV pre-loading disintegrated completely except for PLA_IND.

#### 3.2.3. Gel Permeation Chromatography

GPC analysis revealed changes in M_n_ during the degradation of each sample, with clear differences between the paths with and without UV exposure (Figure 10 and Figure 11). The spinning process had the most significant impact on the M_n_ of the PBS samples, but the effect was generally weak in the PLA samples, with only PLA_SF showing a significant decline. In contrast, the M_n_ of PLA samples declined substantially during disintegration, whereas there was only a limited further impact on the PBS samples, again with the exception of PLA_SF. A slight increase in M_n_ was observed for the PBS samples during the disintegration trial without UV exposure (Figure 10), and this remained the case for PBS_IND even with UV pre-loading (Figure 11). The M_n_ of all PLA samples declined during UV exposure (300h) and continued to decline in the subsequent disintegration (300h + ISO).

In both degradation paths, the PLA_GEO sample behaved similarly to the other PLA samples. If the same PLA_Granulate is used as a reference, the change from the spinning process alone was similar to PLA_LAB, and the sharp decline during ISO disintegration was similar to PLA_IND. If the PLA_GEO sample was pre-exposed to UV, the decrease in M_n_ was less steep compared to the other PLA samples. The M_n_ following ISO disintegration was almost identical for PLA_GEO and PLA_IND. 

The calculated K values are listed in Table 7 for path 2.1 and Table 8 for path 2.2. The calculated impact was largest for PLA_IND_ISO (K = 18.842), followed by PLA_GEO_ISO (K = 11.336), PBS_IND_300h (K = 14.47), PBS_LAB_300h + ISO (K = 10.72), and PLA_GEO_300h + ISO (K = 10). As stated earlier, the PLA_LAB and PLA_SF samples were completely degraded in the disintegration trials with and without UV pre-loading. The PBS_SF sample was insoluble in HFIP (which worked for the other PBS samples) as well as chloroform, tetrahydrofuran, and dimethylformamide. 

## 4. Discussion

### 4.1. Fiber Properties

The properties of spun PLA and PBS fibers (summarized in Table 2) reveal that both polymers are comparable at different spinning scales, as shown by the similar dwell times, draw ratios, and nozzle counts, although the draw ratios differ slightly due to the optimization of the spinning runs. The dwell time, draw ratio, and throughput are almost identical when comparing the two polymers at a given scale. The titer, tensile strength, elongation, and crystallinity differ more significantly between PLA and PBS, reflecting the material characteristics of each polymer. PLA has a higher tenacity, lower elongation at break, lower crystallinity, and lower titer than PBS. With the exception of PBS_LAB, the PBS fibers were larger in diameter, lower in tenacity, higher in elongation, and higher in crystallinity. Dwell time is one comparative parameter that has been used in previous studies [16], and we found that it increases from LAB to IND to SF (pilot) scales.

Notably, PBS_LAB fibers were similar in diameter to PLA fibers and significantly more tenacious than other PBS samples. In contrast, the PLA fibers demonstrated the lowest titer (<4 dtex), whereas PBS_LAB and PBS_SF had titers of ~4 dtex, and PBS_IND exceeded 9 dtex. There is less space for the molecular chains in narrower fibers, making the fiber diameter a critical production parameter. Finer fibers are more likely to break due to irregularities in the polymer melt [42]. We therefore selected a larger fiber diameter to ensure a more stable production run for PBS_IND.

PLA fibers were generally more tenacious than PBS fibers, with a lower elongation, crystallinity, and titer. The greater elongation of PBS fibers suggests a potential for further process optimization to align the polymer chains and increase tenacity [42]. However, the higher crystallinity of PBS fibers implies a limited scope for the polymer chains to align, which should be addressed in future research.

### 4.2. Change Due to Degradation

The comparative DSC analysis of the two degradation paths (Figure 2) showed a significant difference between the polymers (Figure 4 and Figure 5). For PLA, the enthalpy and location of the melt peak changed for each sample, whereas only the enthalpy changed for the PBS samples. This revealed a change in T_m_ for PLA but not PBS. A loss of crystallinity was observed in the PBS_LAB and PBS_IND samples during processing, whereas the crystallinity increased significantly during ISO disintegration (Figure 6). The crystallinity of PBS_SF increased slightly, but further change was marginal during the disintegration trial. The crystallinity of PLA_SF decreased during processing, whereas that of PLA_IND increased to almost 50%, but PLA_LAB and PLA_SF disintegrated completely during ISO degradation, hence the missing data points for the ISO (Figure 6) and 300h + ISO (Figure 7) measurements. When the samples were pre-exposed to UV, the impact of disintegration was more significant (Figure 7). For PLA_IND, the crystallinity and T_m_ did not change much after disintegration. A change in T_m_ from 162 to 152 °C was only observed for PLA_IND (Figure 8 and Figure 9). For PBS_SF and PBS_LAB, the crystallinity after disintegration was almost 10% higher when pre-exposed to UV (Figure 7). With increasing crystallinity during degradation, the amorphous structures of the sample are degraded first due to their accessibility.

The FTIR data revealed a significant change in CI for PBS when the granular sample was processed into fibers (Table 4), as well as changes in CI and VI (Table 6) when PLA degraded without UV exposure. There were also small changes in HI (Table 5). The observation of carbonyl, vinyl, and hydroxyl groups as degradation products confirmed the degradation of the PLA and PBS samples [32,33].

A significant change in M_n_ was observed for all samples during processing and degradation (Figure 10 and Figure 11). For PBS, the most significant change occurred during spinning but little further change occurred during degradation. PLA showed the opposite profile, with little effect during spinning but a larger change during degradation [16]. These results indicate that PBS was more stable than PLA during the degradation tests, and this was confirmed by the complete disintegration of two PLA samples. 

Changes in physical properties (T_m_, crystallinity, and M_n_) were observed in previous studies and were linked to the time the polymer was exposed to higher temperatures, which led to chain cleavage and changes in mechanical behavior [16,29,43]. The dwell time may therefore have a discrete effect on the polymer in addition to the shear stress in the extruder. Chain cleavage in the PLA_IND sample could be responsible for the lower M_n_ and increase in crystallinity. The insolubility of the PBS_SF samples may reflect further polymerization during the extrusion process, with longer dwell times reducing the solubility of the samples [44]. This in turn affects the crystallinity, which did not change substantially for the PBS_SF sample during degradation. Indeed, PBS_SF was the only sample that steadily increased in crystallinity during the course of degradation, without a drop during the spinning process, ultimately reaching a value exceeding 70%.

The findings of this study are summarized in Table 9 for PLA and Table 10 for PBS, revealing connections between different processing parameters and fiber properties. For PLA, we observed a clear connection between higher dwell times, lower fiber diameters, and UV pre-exposure, all of which significantly accelerated the degradation of the sample. The PLA samples with the highest dwell time and the smallest diameter were completely disintegrated during the tests. None of the PBS samples fully disintegrated, and the relationships we observed only indicate tendencies. PBS_SF was insoluble, reflecting the longer dwell time and the resulting increase in polymerization, which means that a longer dwell time cannot, in this case, be linked to faster degradation. PBS overall showed a higher crystallinity, increasing its stability in the disintegration trials. A smaller diameter, which was connected to higher tenacity and lower elongation, also increased the surface area of the fiber relative to the mass, thus accelerating its degradation. UV exposure did not significantly affect the degradation of PBS, in contrast to the effect of UV on PLA fibers.

## 5. Conclusions

A connection between longer dwell times and a change in mechanical properties was established for PLA but was only indicated for PBS because the PBS sample with the longest dwell time was insoluble and could not be characterized by GPC. The connection between a longer dwell time and lower T_m_ was established for PLA. A longer dwell time was also connected to a lower M_n_ for all samples, in agreement with previous reports [29,43]. The samples increased in crystallinity, and in the case of PBS_SF (with the longest dwell time), this was almost linear from the granular sample along the entire degradation path. All other (PBS and PLA) samples decreased in crystallinity after spinning. Increasing crystallinity is linked to the degradation of amorphous structures first. For PLA, the samples with the longest dwell times showed the lowest crystallinity and M_n_ values. 

UV pre-exposure had a strong effect on the disintegration of the PLA and PBS samples, significantly increasing the CI, VI, and crystallinity while slightly reducing the M_n_ compared to samples that were not exposed. This was also found in previous studies, where UV pre-exposure influenced the degradation rate of various polymers [45]. Disintegration is a degradation method that proceeds from the surface inwards. Faster degradation results from the UV radiation changing the surface topography [46]. 

Two of three PLA samples disintegrated completely when maintained at 38 °C for 45 days. Degradation is usually connected to the glass transition temperature (T_g_) of a polymer, meaning that PLA should be degraded at temperatures exceeding 60 °C [4]. We observed the complete disintegration of PLA_LAB and PLA_SF, which was not solely linked to the dwell time of the spinning trial. Indeed, the longer PLA was processed in the extruder (longer dwell time), the lower the crystallinity, the lower the M_n_, and the lower the T_m_, leading to a combinatorial effect that accelerated the degradation. For PBS, a longer dwell time was connected to the highest crystallinity overall in this study. Furthermore, PLA was more stable during processing but was more prone to degradation in the ISO disintegration trial, whereas the opposite was true for PBS.

Interestingly, we found that laboratory-scale fibers (the focus of state-of-the-art studies thus far) behave differently to pre-loaded industrial-scale fibers, representing products placed on the market that end up in the environment. The scale of the spinning process increases the dwell time, which results in shorter polymer chains and faster degradation. If a product is manufactured at the laboratory scale, the findings of any tests may differ significantly from those applied to products from industrial-scale processes. Tests for degradability should therefore accommodate pre-loading that reflects the real-world situation. In our study, quick weathering as a pre-loading method paints a more accurate picture than the testing of pristine fibers, and we recommend this approach for product testing in the future. 

In addition to the dwell time, the textile structure, temperature, and many other parameters affect the degradation process. In this study, the dwell time was isolated as far as possible at three different production scales to show its impact on degradation. Our results suggest that the degradability of biopolymers could be tailored by manipulating the dwell time during production, and this should be validated in further studies with parallel spinning trials involving a wider range of dwell times. Our findings could revolutionize the approach to biopolymer production, leading to more sustainable and environmentally friendly products. The assessment of changes in crystallinity, FTIR-based indices, and the K value determined from GPC data, offer profound insight into the changes that take place in polymer samples as they are broken down.

## Figures and Tables

**Figure 1 polymers-16-01650-f001:**
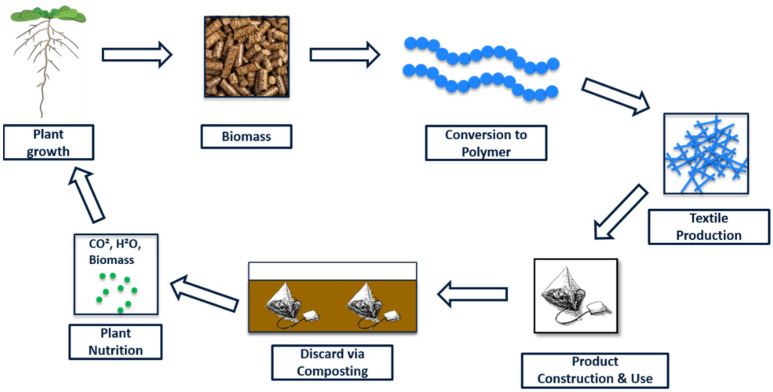
Schematic life cycle of a textile product such as a teabag made from biobased biodegradable polymer fibers.

**Figure 2 polymers-16-01650-f002:**
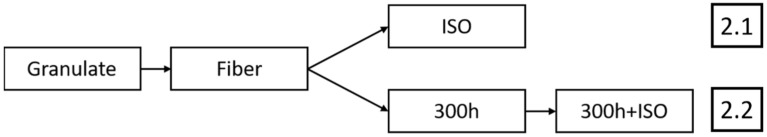
Samples taken during the processing and degradation of two polymers.

**Figure 3 polymers-16-01650-f003:**
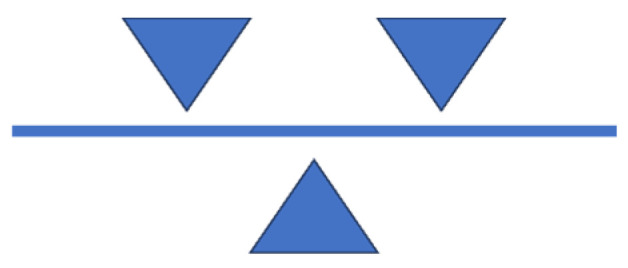
Solidification pattern (needle entrance direction).

**Figure 4 polymers-16-01650-f004:**
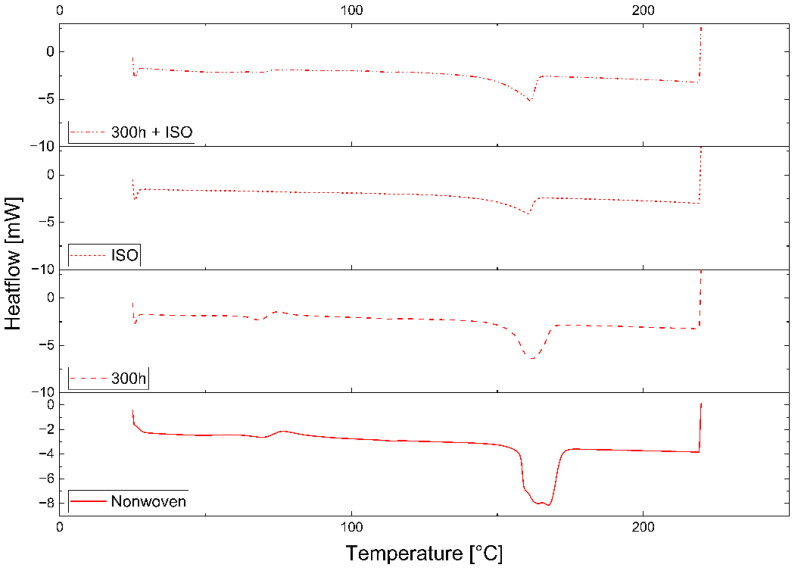
Analysis of PLA_IND by DSC, showing the first heating cycle 25–220 °C: nonwoven (solid), 300h UV (long dash), ISO (short dash), 300h + ISO (dot dash).

**Figure 5 polymers-16-01650-f005:**
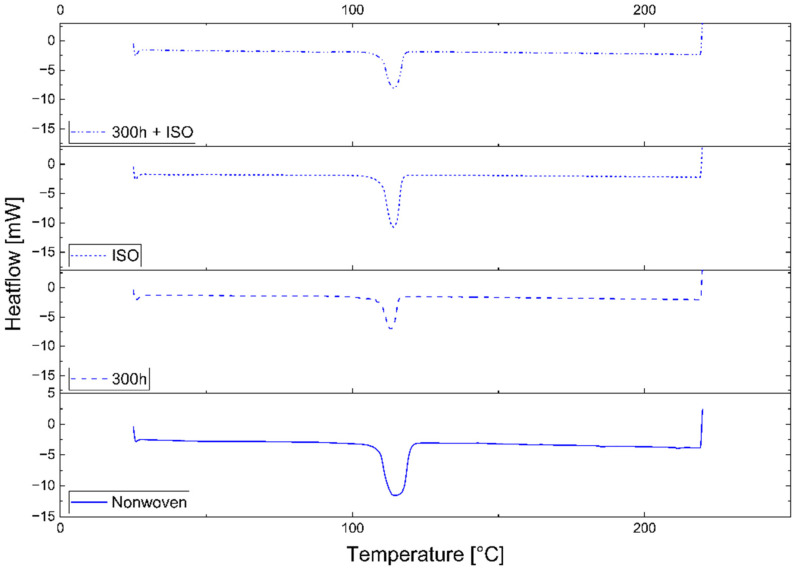
Analysis of PBS_IND by DSC, showing the first heating cycle 25–220 °C: nonwoven (solid), 300h UV (long dash), ISO (short dash), 300h + ISO (dot dash).

**Figure 6 polymers-16-01650-f006:**
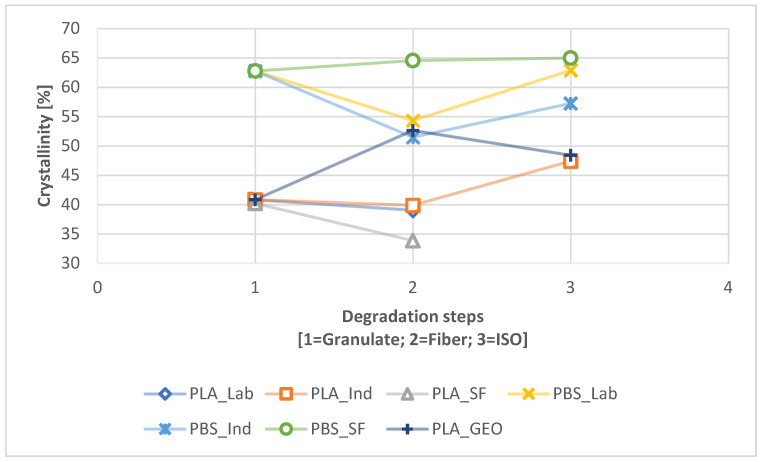
Change in crystallinity during the degradation process from granulate to fiber to ISO (path 2.1).

**Figure 7 polymers-16-01650-f007:**
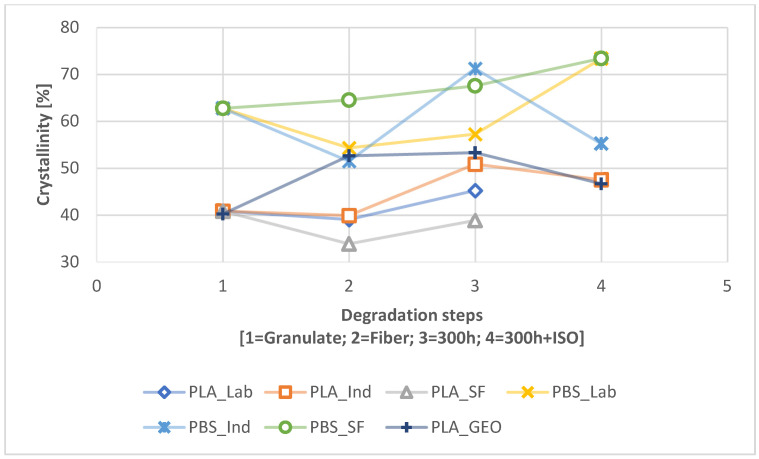
Change in crystallinity during the degradation process from granulate to fiber to 300h to 300h + ISO (path 2.2).

**Figure 8 polymers-16-01650-f008:**
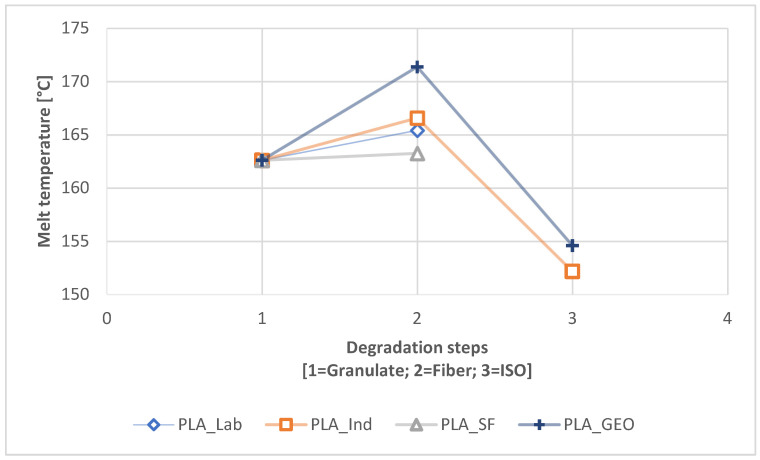
Change in T_m_ during the degradation process from granulate to fiber to ISO (path 2.1).

**Figure 9 polymers-16-01650-f009:**
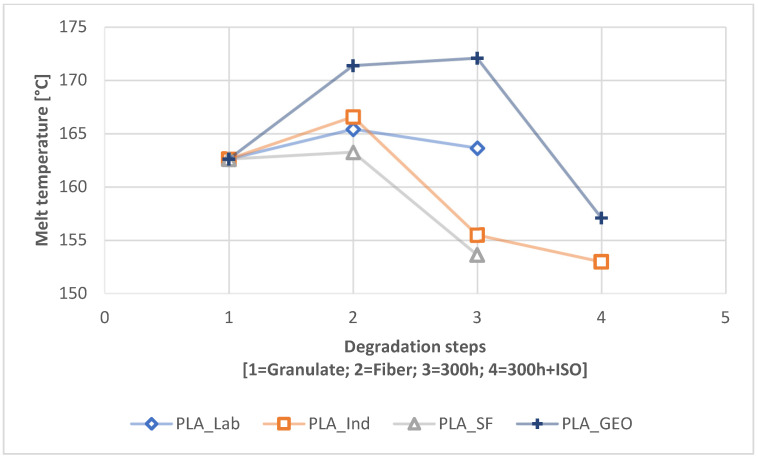
Change in T_m_ during the degradation process from granulate to fiber to 300h to 300h + ISO (path 2.2).

**Figure 10 polymers-16-01650-f010:**
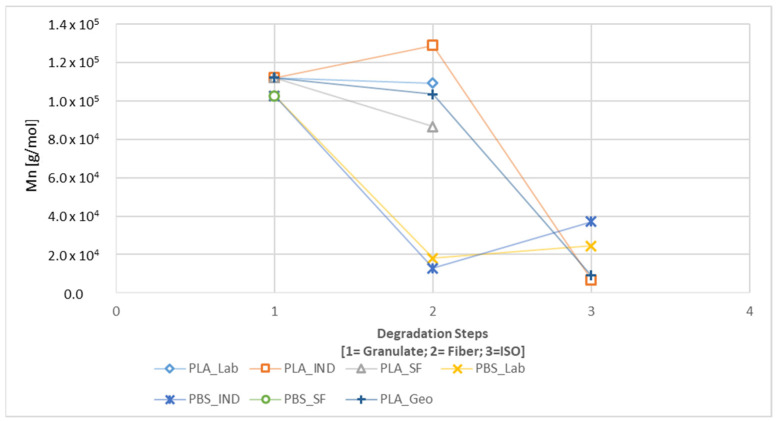
Change in M_n_ during the degradation process from granulate to fiber to ISO (path 2.1).

**Figure 11 polymers-16-01650-f011:**
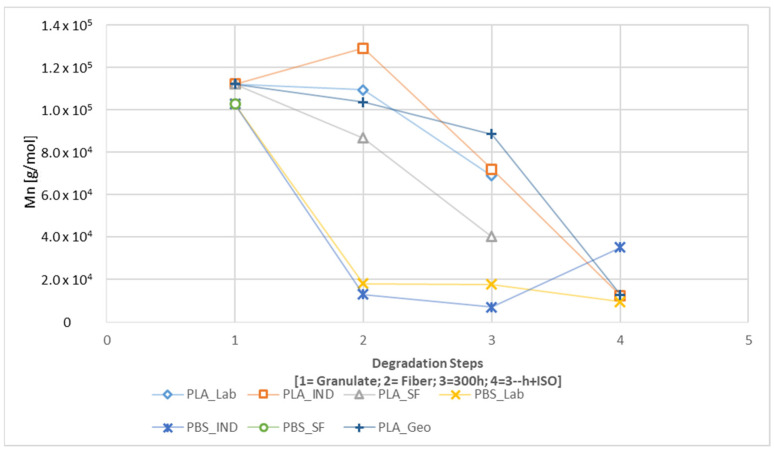
Change in M_n_ during the degradation process from granulate to fiber to 300h to 300h + ISO (path 2.2).

**Table 1 polymers-16-01650-t001:** Parameters associated with different spinning scales.

Spinning Scale	Throughput [kg/h]	Nozzle Count	Dwell Time [min]
PLA_LAB	1	829	5
PLA_IND	368	42.922	7
PLA_SF	30	10.005	36
PBS_LAB	1	829	5
PBS_IND	351	42.922	7
PBS_SF	23	10.005	22

**Table 2 polymers-16-01650-t002:** Machine settings and physical properties. The samples are listed by increasing dwell time.

Sample	Throughput [kg/h]	Dwell Time [min]	Draw Ratio	Titer [dtex]	cN/tex	Elongation [%]	Crystallinity [%]
PLA_LAB	1	5	2.4	3.91	19.28	43.78	39.03
PLA_IND	368	7	1.9	2.6	26.96	45.66	39.88
PLA_SF	30	36	1.82	2.19	23.73	39.55	33.84
PBS_LAB	1	5	3.2	4.18	23.41	116.21	54.3
PBS_IND	351	7	1.75	9.06	15.97	214.74	51.46
PBS_SF	23	22	1.76	4.24	12.86	184.51	64.51

**Table 3 polymers-16-01650-t003:** Changes in crystallinity due to degradation. The pathway without UV exposure (2.1) is highlighted in orange, and the pathway with UV exposure (2.2) is highlighted in blue. No data are available for the PLA_LAB and PLA-SF samples due to their complete disintegration.

Crystallinity	Path	PLA_LAB	PLA_IND	PLA_SF	PBS_LAB	PBS_IND	PBS_SF	PLA_GEO
**Granulate**		40.82	40.82	40.82	62.74	62.74	62.74	40.82
**Fiber**		39.03	39.88	33.84	54.30	51.46	64.51	52.61
**ISO**	2.1	N.A.	47.40	N.A.	62.85	57.21	64.95	48.4
**300h**	2.2	45.22	50.84	38.84	57.23	71.22	67.54	53.29
**300h + ISO**	2.2	N.A.	47.50	N.A.	73.31	55.20	73.38	46.66

**Table 4 polymers-16-01650-t004:** Change in the carbonyl index during degradation. The pathway without UV exposure (2.1) is shown in orange, and the pathway with UV exposure (2.2) is shown in blue.

Carbonyl Index	Path	PLA_GEO	PLA_LAB	PLA_IND	PLA_SF	PBS_LAB	PBS_IND	PBS_SF
**Granulate**		1.94789	1.94789	1.94789	1.94789	5.01155	5.01155	5.01155
**Fiber**		2.55801	7.50089	2.99638	2.23939	7.50089	6.59484	7.38402
**ISO**	2.1	1.52021		1.70042		6.86914	6.24853	6.69622
**300h**	2.2	2.88662	7.59869	2.41957	2.28803	7.59869	6.70654	7.15743
**300h + ISO**	2.2	2.09586		1.99091		7.19465	6.57988	6.86802

**Table 5 polymers-16-01650-t005:** Change in the hydroxyl index during degradation. The pathway without UV exposure (2.1) is shown in orange, and the pathway with UV exposure (2.2) is shown in blue.

Hydroxyl Index	Path	PLA_GEO	PLA_LAB	PLA_IND	PLA_SF	PBS_LAB	PBS_IND	PBS_SF
**Granulate**		0.20621	0.20621	0.20621	0.20621	0.04891	0.04891	0.04891
**Fiber**		0.36899	0.20037	0.35473	0.41445	0.20037	0.21611	0.18229
**ISO**	2.1	0.3296		0.43955		0.24062	0.26277	0.22467
**300h**	2.2	0.46371	0.22207	0.40568	0.40011	0.22207	0.22213	0.20362
**300h + ISO**	2.2	0.42325		0.48282		0.23186	0.22818	0.22155

**Table 6 polymers-16-01650-t006:** Change in the vinyl index during degradation. The pathway without UV exposure (2.1) is shown in orange, and the pathway with UV exposure (2.2) is shown in blue.

Vinyl Index	Path	PLA_GEO	PLA_LAB	PLA_IND	PLA_SF	PBS_LAB	PBS_IND	PBS_SF
**Granulate**		1.78049	1.78049	1.78049	1.78049	2.82858	2.82858	2.82858
**Fiber**		2.33928	2.92216	2.25796	2.1267	2.92216	2.95236	2.91245
**ISO**	2.1	0.93223		0.71412		2.37826	2.37442	2.34196
**300h**	2.2	2.90349	2.93925	2.21447	2.07426	2.93925	3.08622	2.97048
**300h + ISO**	2.2	1		0.98249		2.5889	2.64028	2.50618

**Table 7 polymers-16-01650-t007:** Parameter K over the course of degradation path 2.1. N.A. indicates the absence of data because the samples completely disintegrated or were insoluble.

Sample	Fiber	ISO
PLA_LAB	1.01	N.A.
PLA_IND	0.857	18.842
PLA_SF	1.273	N.A.
PBS_LAB	5.67	4.19
PBS_IND	7.9	2.77
PBS_SF	N.A	N.A.
PLA_GEO	1.236	11.336

**Table 8 polymers-16-01650-t008:** Parameter K over the course of degradation path 2.2. N.A. indicates the absence of data because the samples completely disintegrated or were insoluble.

Sample	Fiber	300h	300h + ISO
PLA_LAB	1.01	1.604	N.A.
PLA_IND	0.857	1.536	8.81
PLA_SF	1.273	2.747	N.A.
PBS_LAB	5.67	5.77	10.72
PBS_IND	7.9	14.47	2.92
PBS_SF	N.A	N.A.	N.A.
PLA_GEO	1.236	1.446	10.000

**Table 9 polymers-16-01650-t009:** Summary of conclusions for PLA. Arrows in the *Change toward* column indicate the direction of change in the parameters listed on the left (↑ = significant increase, ↓ = significant decrease, ↗ = increase, ↘ = decrease, yes = applied). Arrows in the fiber property columns indicate the effect of these parameters. Gray cells indicate effects that were not tested. Green shading indicates process parameters that we controlled.

Parameter	Change Toward	Crystallinity	Tenacity	Elongation	CI	VI	HI	M_n_	Degradability
Dwell time	↑	↓	↘	↘	↗	↗	↔	↓	↑
Draw ratio	↗	↗	↗	↘	↗	↗	↘	↔	↘
Fiber diameter	↘	↗	↗	↓	↗	↘	↗	↔	↑
Tenacity	↗	↗		↘	↗	↗	↘	↔	↘
Elongation	↘	↗	↗		↗	↗	↘	↔	↘
UV exposure before disintegration	yes	↑			↗	↗	↗	↓	↑

**Table 10 polymers-16-01650-t010:** Summary of conclusions for PBS. Arrows in the *Change toward* column indicate the direction of change in the parameters listed on the left (↑ = significant increase, ↓ = significant decrease, ↗ = increase, ↘ = decrease, yes = applied). Arrows in the fiber property columns indicate the effect of these parameters. Gray cells indicate effects that were not tested. Green shading indicates process parameters that we controlled.

Parameter	Change Toward	Crystallinity	Tenacity	Elongation	CI	VI	HI	M_n_	Degradability
Dwell time	↑	↑	↘	↘	↘	↔	↔	↘	↘
Draw ratio	↗	↔	↑	↘	↗	↔	↔	↗	↗
Fiber diameter	↘	↗	↑	↘	↗	↘	↔	↗	↗
Tenacity	↗	↘		↘	↘	↗	↔	↗	↗
Elongation	↘	↘	↗		↘	↗	↔	↗	↗
UV exposure before disintegration	yes	↑			↑	↑	↘	↔	↗

## Data Availability

The data that support the findings of this study are available from the corresponding author G.H. Seide, upon reasonable request due to privacy.

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
