# Peer review of "A Comparison of Laboratory and Industrial Processes Reveals the Effect of Dwell Time and UV Pre-Exposure on the Behavior of Two Polymers in a Disintegration Trial"

_polymers, 2024, doi:10.3390/polym16121650_

Round 1

Reviewer 1 Report

Comments and Suggestions for Authors

This paper compares the degradation of PLA and PBS fibers produced by different methods. The dwell time in the extruder was chosen as the main comparison parameter, as it is the easiest to track. The article undoubtedly deserves publication after some revision:

1) Did the authors consider the possibility of rheological analysis of polymers depending on dwell time? As far as we know from the literature PLA is very sensitive to its thermal history and processing time can affect the rheological properties of the polymer.

2) It is not very clear from the article what PLA_GEO is. This abbreviation has not been introduced anywhere. Sometimes it is mentioned in comparison with PLA_Granulated, but as far as I understand they are different concepts. I realize that there are a lot of abbreviations in the article, and maybe I missed something.

3) page 2 line 76 Wrong index in TiO2 designation.

4) What I find most strange is the very small discussion section and the huge conclusion. In my opinion, the tables and references should be moved from the conclusion to the discussion section, and the conclusion should be shortened to a couple of paragraphs.

5) In graphs where degradation stages are indicated by numbers and their designation is written below, it is possible to sign the name of degradation stages on the axis at once, it should simplify the perception of already complicated graphs. Perhaps the authors can suggest other variants of graphs simplification.

In general, the article is very interesting and addresses an important gap in predicting degradation between laboratory and industrial samples. After revision, it can be published in the journal Polymers

Author Response

#1

Open Review

Quality of English Language

(x) I am not qualified to assess the quality of English in this paper
( ) English very difficult to understand/incomprehensible
( ) Extensive editing of English language required
( ) Moderate editing of English language required
( ) Minor editing of English language required
( ) English language fine. No issues detected

Yes

Can be improved

Must be improved

Not applicable

Does the introduction provide sufficient background and include all relevant references?

(x)

( )

( )

( )

Is the research design appropriate?

( )

(x)

( )

( )

Are the methods adequately described?

( )

(x)

( )

( )

Are the results clearly presented?

( )

(x)

( )

( )

Are the conclusions supported by the results?

( )

( )

(x)

( )

Thank you very much for the comments on our publication. Below, you will find our reply as well, what we have changed.

Comments and Suggestions for Authors

This paper compares the degradation of PLA and PBS fibers produced by different methods. The dwell time in the extruder was chosen as the main comparison parameter, as it is the easiest to track. The article undoubtedly deserves publication after some revision:

1) Did the authors consider the possibility of rheological analysis of polymers depending on dwell time? As far as we know from the literature, PLA is very sensitive to its thermal history, and processing time can affect the polymer's rheological properties.

In rheology, the change in molecular mass causes a change in viscosity, known as melt viscosity. This can also be observed using a GPC measurement. The solution viscosity is determined by placing the sample in solution and testing it in a GPC (Gel Permeation Chromatography), which provides information about the molecular mass change. Both paths lead to the same goal. These two viscosities can be linked.

2) It is not very clear from the article what PLA_GEO is. This abbreviation has not been introduced anywhere. Sometimes it is mentioned in comparison with PLA_Granulated, but as far as I understand they are different concepts. I realize that there are a lot of abbreviations in the article, and maybe I missed something.

PLA_GEO is a Geotextile that has been used as a comparison material. This was added in Chapter 2.1.

3) page 2 line 76 Wrong index in TiO2 designation.

Corrected.

4) What I find most strange is the very small discussion section and the huge conclusion. In my opinion, the tables and references should be moved from the conclusion to the discussion section, and the conclusion should be shortened to a couple of paragraphs.

This was revised.

5) In graphs where degradation stages are indicated by numbers and their designation is written below, it is possible to sign the name of degradation stages on the axis at once, it should simplify the perception of already complicated graphs. Perhaps the authors can suggest other variants of graphs simplification.

Due to the long names of the different degradation stages, this labeling of the x-axis was chosen to improve the graph overview.

In general, the article is very interesting and addresses an important gap in predicting degradation between laboratory and industrial samples. After revision, it can be published in the journal Polymers

Submission Date

06 May 2024

Date of this review

14 May 2024 14:05:44

Reviewer 2 Report

Comments and Suggestions for Authors

 2-8- According to standards, the biodegradation criterion of biopolymer is CO2 emission (e.g. ISO17556).

 Therefore, disintegration isn’t a suitable criterion for evaluation.

2-8- In this section, which phenomena was studied? the disintegration of fragmentation? (Standard EN 13432)

3-2-1- Degradation is a process involved in producing small molecules, volatile, chain session. 

What is the mechanism of crystalline change in the granule-to-fiber process during degradation?

In this paper, degradation and biodegradation are discussed. What is the primary goal of the study?

The paper has no clear and valid evidence for biodegradation phenomena.

Typo error:

-        Line:109

-        Line:151

-title is long and must be short

  Comments on the Quality of English Language

some items must be correct

Author Response

#2

Open Review

Quality of English Language

( ) I am not qualified to assess the quality of English in this paper
( ) English very difficult to understand/incomprehensible
( ) Extensive editing of English language required
( ) Moderate editing of English language required
(x) Minor editing of English language required
( ) English language fine. No issues detected

Yes

Can be improved

Must be improved

Not applicable

Does the introduction provide sufficient background and include all relevant references?

(x)

( )

( )

( )

Is the research design appropriate?

( )

(x)

( )

( )

Are the methods adequately described?

( )

(x)

( )

( )

Are the results clearly presented?

(x)

( )

( )

( )

Are the conclusions supported by the results?

(x)

( )

( )

( )

Thank you very much for the comments on our publication. Below, you will find our reply as well, what we have changed.

Comments and Suggestions for Authors

  • 2-8- According to standards, the biodegradation criterion of biopolymer is CO2 emission (e.g. ISO17556).

            Therefore, disintegration isn’t a suitable criterion for evaluation.

According to DOI 10.1007/s10924-011-0317-1, the different phenomena of polymer degradation happen sequentially or simultaneously. The mechanisms at work (https://doi.org/10.1007/s42452-021-04851-7) are from start to finish

  • deconstruction (disassembly into macroscopic component parts)
  • disintegration (reduction in physical size (<10 mm); this occurs by chemical action and microorganisms rather than mechanical force)
  • enzymatic catalysis (breaking down of the chemical structure with enzymes)
  • assimilation (the uptake/ incorporation of the material into cells)
  • bioresorption (Degradation due to the contact with the environment that leads to complete assimilation)

It seems that the term "disintegration" may not accurately convey the mechanism at hand. However, it is worth noting that this term is widely utilized in other publications, which is why it has also been included here.

  • 2-8- In this section, which phenomena was studied? the disintegration of fragmentation? (Standard EN 13432)

Disintegration was studied based on ISO 20200 (“Determination of the degree of disintegration of plastic materials under simulated composting conditions in a laboratory-scale test”).

Thank you for mentioning the EN 13432 standard. However, this standard doesn't cover disintegration; it's actually related to packaging. This study mainly deals with fibers produced by three production machines, not so much the later application. The ISO 20200 standard aligns more closely with the study's goals.

  • 3-2-1- Degradation is a process involved in producing small molecules, volatile, chain session. 

3.2.1 focuses on the DSC measurements performed.

  • What is the mechanism of crystalline change in the granule-to-fiber process during degradation?

The question can relate to two things.

1) on the change in crystallinity in the spinning process from granules to fiber or

2) on the change in crystallinity in the degradation process itself.

  • Firstly, there is the orientation of the molecular chains, which are oriented in the spinning process. This leads to an increase in crystallinity.
  • Secondly, the accessibility of the amorphous areas of the fiber plays a pivotal role. Their easier access makes them more susceptible to degradation, leading to a significant increase in crystallinity due to the degradation induced change in molecular weight.

  • In this paper, degradation and biodegradation are discussed. What is the primary goal of the study?

The initial comment above provides a more detailed explanation of the distinction between disintegration and biodegradation. This study specifically focuses on disintegration, referring to the sample's disintegration and the resulting increase in surface area. This increase in surface area (disintegration) facilitates improved accessibility for subsequent biodegradation.

  • The paper has no clear and valid evidence for biodegradation phenomena.

The distinction between disintegration and biodegradation is crucial to understanding the processes involved. This study delves into the concept of disintegration, which involves the breakdown of the sample and the subsequent increase in surface area. The resulting heightened surface area, plays a pivotal role in enhancing accessibility for subsequent biodegradation.

  • Typo error:

-            Line:109

-            Line:151

-           title is long and must be short 

  • Comments on the Quality of English Language

some items must be correct

The English language has been proofread by a professional native speaking proofreader.

Submission Date

06 May 2024

Date of this review

23 May 2024 07:55:25

Reviewer 3 Report

Comments and Suggestions for Authors In the work the effect of dwell time and UV pre-exposure on the behavior of PLA and PBS of laboratory and industrial processes were revealed. The work can be considered for publication after somemodifications.
(1)FTIR were used to determined the chemical profile of the granulated polymer, fibers before and 132 after UV exposure, and fibers following the disintegration. It was calculated by the heght of peak. I surggest use area of peak to calculate.
(2)How the Crystallinity in table 2 obtained should be noted.
(3)Figure 4 and Figure 5 shoule improved becaused the curves were not clear.
(4)Typical FTIR figure should be shown. Comments on the Quality of English Language

Minor editing of English language required.

Author Response

#3

Open Review

Quality of English Language

( ) I am not qualified to assess the quality of English in this paper
( ) English very difficult to understand/incomprehensible
( ) Extensive editing of English language required
( ) Moderate editing of English language required
(x) Minor editing of English language required
( ) English language fine. No issues detected

Yes

Can be improved

Must be improved

Not applicable

Does the introduction provide sufficient background and include all relevant references?

(x)

( )

( )

( )

Is the research design appropriate?

( )

(x)

( )

( )

Are the methods adequately described?

( )

(x)

( )

( )

Are the results clearly presented?

(x)

( )

( )

( )

Are the conclusions supported by the results?

(x)

( )

( )

( )

Thank you very much for the comments on our publication. Below, you will find our reply as well, what we have changed.

Comments and Suggestions for Authors

In the work the effect of dwell time and UV pre-exposure on the behavior of PLA and PBS of laboratory and industrial processes were revealed. The work can be considered for publication after somemodifications.

(1)FTIR were used to determined the chemical profile of the granulated polymer, fibers before and 132 after UV exposure, and fibers following the disintegration. It was calculated by the heght of peak. I surggest use area of peak to calculate.

If carbonyl, hydroxyl, and vinyl groups are compared (as in the literature mentioned 33-35), the intensity is the comparison parameter. The arithmetic mean was used in the comparison when considering the ranges for carbonyl (1710-1713) and hydroxyl (3423-3429). The same was done for all samples. The exact distance between the frequencies (1710-1713 and 3423-3429) and the arithmetic mean calculation depict the same accurate results as the area.

(2)How the Crystallinity in table 2 obtained should be noted.

All crystallinity measurements have been obtained with the DSC measurements described in 2.2 and calculated with Equation 1. The melt enthalpy for the 100% crystalline PLA and PBS have been obtained from the literature.

(3)Figure 4 and Figure 5 shoule improved becaused the curves were not clear.

Figures 4 and 5 display legible labels and uniform scaling, and each degradation step has a uniform line representation for easy comparison. The differences between the individual degradation steps (glass transition area for PLA and the melting area of PLA and PBS) are visible. “Not Clear” is not apparent in this context.

(4)Typical FTIR figure should be shown.

The general FTIR curve of a polymer does not fundamentally change during degradation; it only changes in individual areas. Due to the many data points, the curves largely overlap (see below). Listing the FTIR data makes it easier to follow the chain of reasoning.

Comments on the Quality of English Language

Minor editing of English language required.

The English language has been proofread by a professional native speaking proofreader.

Submission Date

06 May 2024

Date of this review

24 May 2024 03:24:40

Round 2

Reviewer 1 Report

Comments and Suggestions for Authors

The authors corrected the article according to my comments, and I received reasonable answers for others. The article can be accepted for printing.

There are two small comments:

1) I never found a mention of PLA_GEO in chapter 2.1, although the authors said they added this explanation. The first time PLA_GEO is mentioned is in section 3.2.

2) Figure 11: A typo in the degradation stages “4=3--h+ISO

After these corrections, the paper will be ready for printing.

Author Response

The errors are corrected.

Reviewer 3 Report

Comments and Suggestions for Authors

I persits that author should revised manuscript according to my comments. 

Comments on the Quality of English Language

Minor editing of English language required

Author Response

The publication was proofread again by our native-speaking professional proofreader, and minor English corrections have been made.

Round 3

Reviewer 3 Report

Comments and Suggestions for Authors

The work can be considered for publication